# A general hypergraph learning algorithm for drug multi-task predictions in micro-to-macro biomedical networks

Shuting Jin [1,2,3], Yue Hong [2], Li Zeng [3], Yinghui Jiang [2], Yuan Lin [4], Leyi Wei [5], Zhuohang Yu [6], Xiangxiang Zeng [7]*, Xiangrong Liu [2,8]*

**1** School of Computer Science and Technology, Wuhan University of Science and Technology, Wuhan, China, **2** School of Informatics, Xiamen University, Xiamen, China, **3** Department of AIDD, Shanghai Yuyao Biotechnology Co., Ltd., Shanghai, China, **4** School of Economics, Innovation, and Technology, Kristiania University College, Bergen, Norway, **5** School of Software, Shandong University, Shandong, China, **6** Shanghai Frontiers Science Center of Optogenetic Techniques for Cell Metabolism, School of Pharmacy, East China University of Science and Technology, Shanghai, China, **7** School of Information Science and Engineering, Hunan University, Hunan, China, **8** Zhejiang Lab, Hangzhou, China

* xzeng@foxmail.com (XZ); xrliu@xmu.edu.cn (XL)

**Data Availability Statement:** Our data and code are available on github online at https://github.com/stjin-XMU/HGDrug.

## Abstract

The powerful combination of large-scale drug-related interaction networks and deep learning provides new opportunities for accelerating the process of drug discovery. However, chemical structures that play an important role in drug properties and high-order relations that involve a greater number of nodes are not tackled in current biomedical networks. In this study, we present a general hypergraph learning framework, which introduces **D**rug-**S**ubstructures relationship into **M**olecular interaction **N**etworks to construct the micro-to-macro drug centric heterogeneous network (**DSMN**), and develop a multi-branches **H**yper-**G**raph learning model, called **HGDrug**, for **Drug** multi-task predictions. HGDrug achieves highly accurate and robust predictions on 4 benchmark tasks (drug-drug, drug-target, drug-disease, and drug-side-effect interactions), outperforming 8 state-of-the-art task specific models and 6 general-purpose conventional models. Experiments analysis verifies the effectiveness and rationality of the HGDrug model architecture as well as the multi-branches setup, and demonstrates that HGDrug is able to capture the relations between drugs associated with the same functional groups. In addition, our proposed drug-substructure interaction networks can help improve the performance of existing network models for drug-related prediction tasks.

## Author summary

Drugs containing the same functional groups may have similar pharmacochemical properties. However, how to effectively combine chemical information of drugs from molecular fragments containing functional groups into the biomolecular network is challenging and rarely explored. we decompose drugs' SMILES string and construct a drug-centric heterogeneous network that integrates drug substructure and molecular interactions

**Funding:** This work was supported by the National Natural Science Foundation of China (grant nos. 62072384, 61872309, 62072385, 61772441 to XL), the Zhijiang Lab (2022RD0AB02 to XL). The funders had no role in study design, data collection and analysis, decision to publish, or preparation of the manuscript.

**Competing interests:** The authors have declared that no competing interests exist.

information. Based on the heterogeneous network, we proposed an end-to-end hypergraph attention network framework for the drug multi-task predictions, termed as HGDrug. The efficiency and generalization of the proposed HGDrug have been demonstrated by the state-of-the-art performance in four drug-related interaction predictions tasks with huge improvement compared to previous general-purpose classical models and task-specific models. In addition, HGDrug can effectively identify potential drug-related interactions and the drug-sub-structure networks are able to help to improve the performance of other GNN models. These conclusions present important insights on how to introduce the drug' substructure information for multiple drug-related interactions tasks on biomedical networks. In summary, HGDrug offers a general and powerful tool for the identification of drug-related interactions by constructing the micro-to-macro drug-centric heterogeneous network.

This is a *PLOS Computational Biology* Benchmarking paper.

## Introductions

Drug design and development are important research areas for pharmaceutical companies and chemical scientists, yet traditional new drug R&D (research and development) costs an average of over $1 billion and can take up to 12 years or more, while still yielding low success rates [1]. With the development of Deep Learning (DL) techniques and the acccessible availabilities of various data resources, there is now a greater interest in applying computational methodologies to expedite drug discovery and development processes. Among them, the graph neural networks (GNNs) have become a powerful tool to promote the drug development process as they can model the structural information of drug molecules and the interactive relationships between biomolecules, therefore attracting growing interest in drug design and development process.

Drugs can exploit interaction with biomolecules and may act on specific targets and proteins in order to treat the disease. Therefore, identifying the interactions between drug-target and drug-disease is of great significance for drug development. At the same time, the interaction between drug-target or drug-disease also provides a higher-level point of view for better understanding the drug-side effects and drug-drug associations [3]. In recent years, drug-related molecular interaction networks, e.g., drug-drug interactions (DDIs), drug-target interactions (DTIs), drug-disease interactions (DDiIs), and drug-side-effect interactions (DSIs) are continuously expanded, which have greatly increased the computational power used to aid drug discovery (Table A in S1 Text displays the abbreviation list in this study for ease of reading). However, there is a great number of undiscovered associations in the existing drug-related interaction networks. An increasing number of GNN models have been proposed to explore these unknown associations. The initial GNN models for the prediction of molecular interactions take only network data into account [4–7]. As available network information is continuously enlarged, researchers have also begun to integrate a growing number of related interaction networks to explore more effective features of the biomolecular nodes [8–11]. Yue et al. [12] select 11 representative graph embedding methods and conduct a systematic comparison on 3 important biomedical link prediction tasks and 2 node classification tasks. In

addition, more research efforts have been put on including more attributes or meta information besides network data, such as the combination of network and text information for the prediction of association between drugs and diseases [13]. The drugs' sequence or graph structure information are also used to improve the performance of drug-related interactions prediction [14, 15]. However, we have observed that most existing network models have certain drawbacks. For example, methods proposed in [13, 16, 17] require a separate model to learn the drug's information, leading to an increased complexity in the overall model structure. Moreover, the non-end-to-end method can degrade the prediction performance since it fails to optimize the learned features according to downstream tasks. On the other hand, chemical structure that play important role in drug properties is neglected in current biomedical networks. Liu et al. [18] present a subcomponent-guided deep learning method for accurate and interpretable CDR prediction, named SubCDR, to recognize the most relevant subcomponents driving response outcomes. However, this method only considers the substructure information of the drug without combining this substructure with biomedical network information. Moreover, conventional network models rely too much on pairwise links and cannot reveal high-order interactions between nodes with similar topology of the neighborhoods, therefore not able to capture the chemical structure information related to the drugs.

To address the aforementioned problems and capture the higher-order information of drugs associated with the same functional groups or molecular interactions, we introduce drugs' substructure (functional group fragments) into biomedical networks and construct the micro-to-macro drug-centric heterogeneous network (DSMN) in this work (Fig 1). Specifically, we design a method to construct drug-substructures interaction networks, which utilize the BRICS (breaking of retrosynthetically interesting chemical substructures) [2] algorithm to decompose

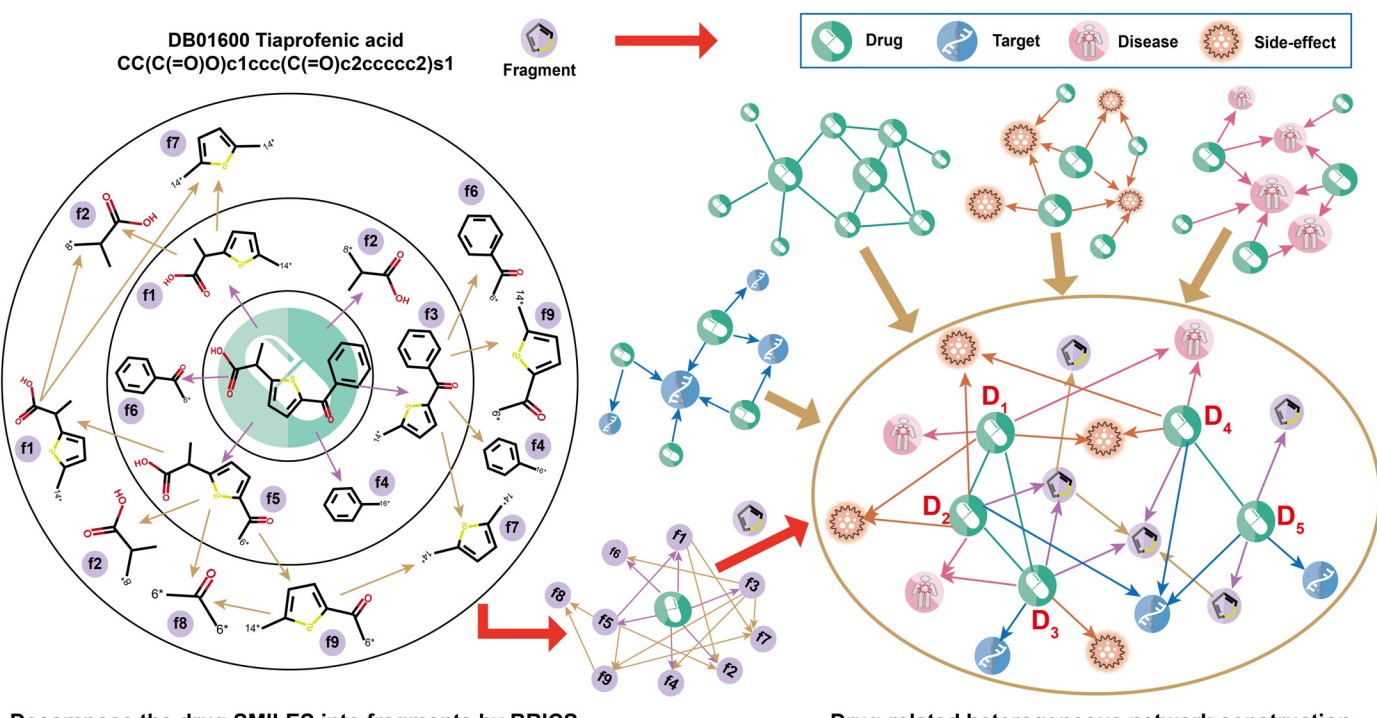

**Fig 1. A micro-to-macro drug-centric heterogeneous network DSMN construction.** An example of segmenting drug SMILES using the BRICS [2] rules to construct the drug-substructure network, and the heterogeneous network DSMN is reconstructed by assembling all drug-related networks.

drugs' SMILES string and obtain all functional group fragments contained in a drug. We construct drug-fragment and fragment-fragment interaction networks with the functional group fragments as nodes, and fuse these drug-substructures networks and molecular interaction networks to construct a micro-to-macro drug-centric heterogeneous network. As hypergraph [19] generalizes the concept of edge to make it connect an arbitrary number of nodes, it provides a natural way to model complex high-order relations among drugs. We then propose a novel hypergraph-learning framework for the prediction of multiple drug-related interactions (termed as HGDrug) based on the DSMN (Fig 2). We carefully design triangular and quadrilateral motifs with underlying semantics to construct hypergraphs, and define multiple categories of motifs that formulate different types of high-order relations, e.g.,"having same chemical structures" or "having same molecular interactions" between drugs depending on whether they are directly related or not. To fully inherit the rich information present in the motif-driven hypergraphs, we incorporated a self-supervised task [20] into the training of the multi-branches hypergraph attention network. In summary, HGDrug exploits the multi-branches hypergraph attention network as the feature encoders of different motifs-driven hypergraphs and uses the self-supervised auxiliary task to avoid the loss of high-order information when aggregating multiple branches drug embeddings to identify potential drug-related interactions.

We conduct extensive experiments to compare our model with 8 state-of-the-art task specific models and 6 general-purpose conventional methods. On the 4 drug-related interactions prediction tasks (Table 1), namely, DDIs, DTIs, DDiIs, and DSIs, the performance of HGDrug is significantly improved compared to the state-of-the-art task specific models (Fig 3). HGDrug achieves also an average of 7.18% and 7.78% improvement in AUROC (the area under the receiver operating characteristic curve), and AUPR (the area under the precision-recall curve) over the second best result of the general methods, respectively as depicted in Table 2. We investigate the effectiveness of multi-branches hypergraph and neural network architectures through ablation study and illustrate that our proposed drug-substructure networks can be used to improve the performance of existing drug-related interaction prediction GNN models (Figs 3 and 4). The biomedical dataset records of our predictions support that HGDrug excels at predicting potential interactions (Table 3). The main contributions of this paper are summarized as follows:

- We propose the micro-to-macro drug-centric heterogeneous network (DSMN) construction method, which introduces drugs' substructure (functional group fragments) into biomedical networks.

- We design drug-related motifs with latent semantics to construct hypergraphs and develop a multi-branches hypergraph learning framework, which can efficiently capture the high-order relations between drugs with important local structural features.

- We perform extensive experiments to compare our model with the state-of-the-art task specific and general models, and prove our model's effectiveness on multiple drug interaction predictions. The drug-substructure network can also help improve the performance of other existing GNN models, which provides a new perspective for network model-based drug discovery.

## Related work

### Heterogeneous graph

In recent years, multi-omics technologies and approaches to systems biology have created large heterogenized biological networks that provide significant opportunities for graph neural

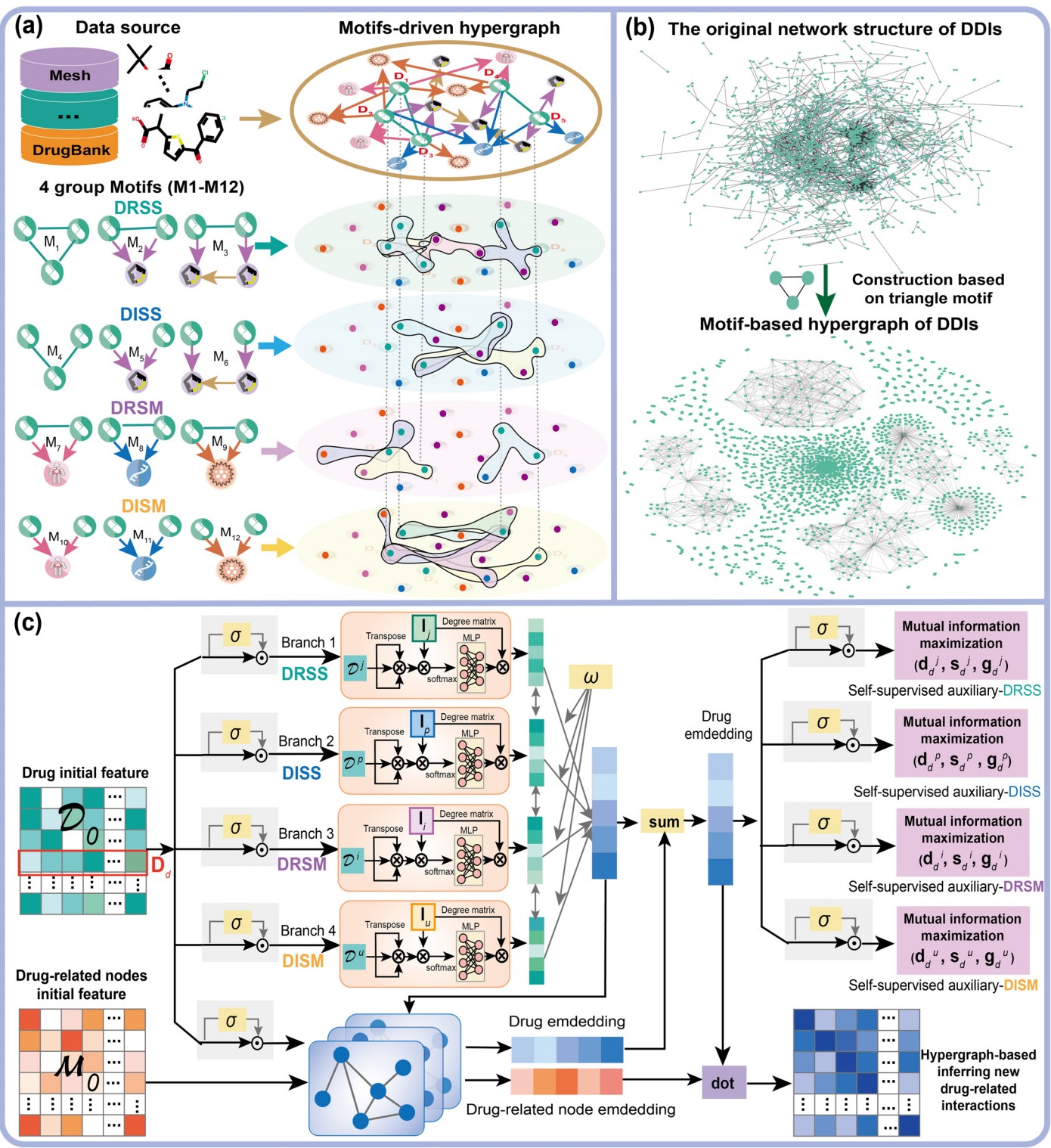

**Fig 2. The framework of HGDrug.** The sub-figure **(a)** depicts all the motifs presented in our work. DRSS($\mathbf{I}_j$), DISS($\mathbf{I}_p$), DRSM ($\mathbf{I}_i$), DISM($\mathbf{I}_u$) denote the four motif-driven hypergraphs constructed on *drug related and have the same substructure*, *drug independent and have the same substructure*, *drug related and have the same molecular interactions*, and *drug independent and have the same molecular interactions* motifs groups, respectively. The sub-figure **(b)** is a real example of driving the hypergraph base on triangular motif. It shows that the hypergraph method will make the relation between some nodes with high-order relations closer. The sub-figure **(c)** draws the process of multi-branches hypergraph attention and graph convolutional networks inferring new drug-related interactions.

**Table 1. Statistical information for the four interactions datasets: The number of drugs, drug-related nodes, fragments and interaction pairs.**

| Dataset | #Node 1 | #Node 2 | #Edges | #Fragments | #DFI | #FFI |
|---|---|---|---|---|---|---|
| Drug-drug | 1,514 | 1,514 | 48,514 | 15,032 | 7,993 | 83,811 |
| Drug-target | 5,017 | 2,324 | 15,138 | 46,855 | 28,453 | 816,918 |
| Drug-disease | 1,519 | 1,229 | 6,677 | 12,982 | 7,509 | 70,574 |
| Drug-side-effect | 1,223 | 5,734 | 153,663 | 11,135 | 6,330 | 58,083 |

networks (GNN) to accelerate the progress of drug discovery. The heterogeneous graph-based GNN methods are used to predict various potential interactions related to drugs. Drug-drug interactions (DDIs) and drug-side-effect interactions (DSIs) predictions can help pharmacologists discover the potential combinations of drugs and the possible side-effects. Zhao et al. [21] use a graph attention network (GAT) to integrate three different features for DSIs prediction. Yu et al. [22] propose a hybrid GNN framework composed of graph and node embedding to

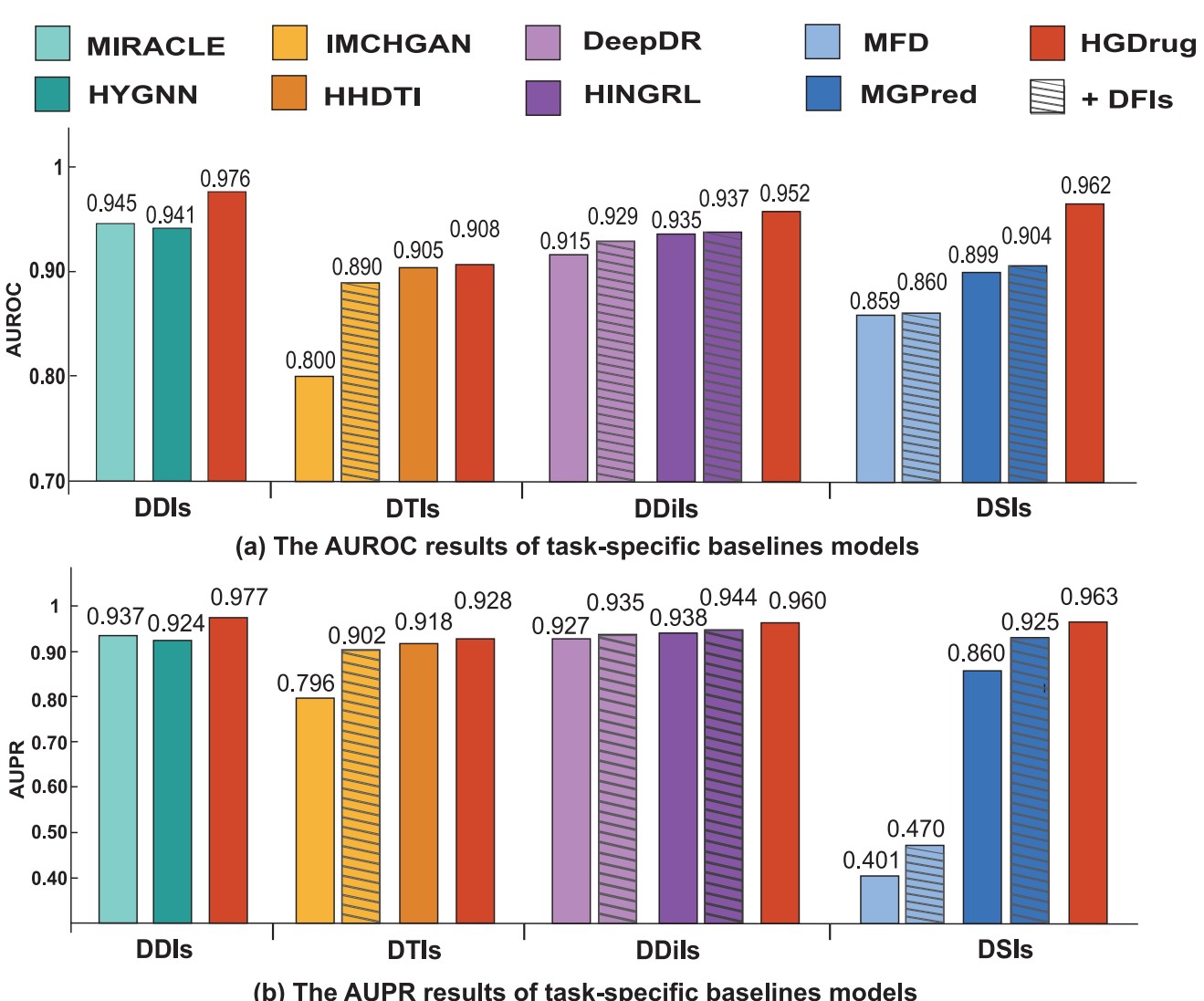

(a) The AUROC results of task-specific baselines models

(b) The AUPR results of task-specific baselines models

**Fig 3. Prediction results of HGDrug and tasks-specific baseline models on the four tasks.**

**Table 2. Prediction results of HGDrug and general baselines models on 4 drug-interactions interactions datasets.** The best performance is marked in bold and the second best is underlined to facilitate reading.

| Model | DDIs | | DTIs | | DDiIs | | DSIs | |
|---|---|---|---|---|---|---|---|---|
| | AUROC | AUPR | AUROC | AUPR | AUROC | AUPR | AUROC | AUPR |
| SVM | 0.613 | 0.591 | 0.670 | 0.621 | 0.602 | 0.580 | 0.655 | 0.603 |
| Katz | 0.665 | 0.643 | 0.672 | 0.644 | 0.701 | 0.722 | 0.750 | 0.732 |
| Deepwalk | 0.722 | 0.701 | 0.723 | 0.759 | 0.801 | 0.799 | 0.852 | 0.833 |
| GCN | 0.858 | 0.830 | 0.883 | 0.888 | 0.839 | 0.861 | 0.929 | 0.934 |
| GAT | 0.831 | 0.791 | 0.840 | 0.851 | 0.818 | 0.828 | 0.931 | 0.932 |
| SkipGNN | 0.858 | 0.834 | 0.839 | 0.856 | 0.811 | 0.839 | 0.929 | 0.932 |
| HGDrug | **0.976** | **0.977** | **0.908** | **0.928** | **0.952** | **0.960** | **0.962** | **0.963** |

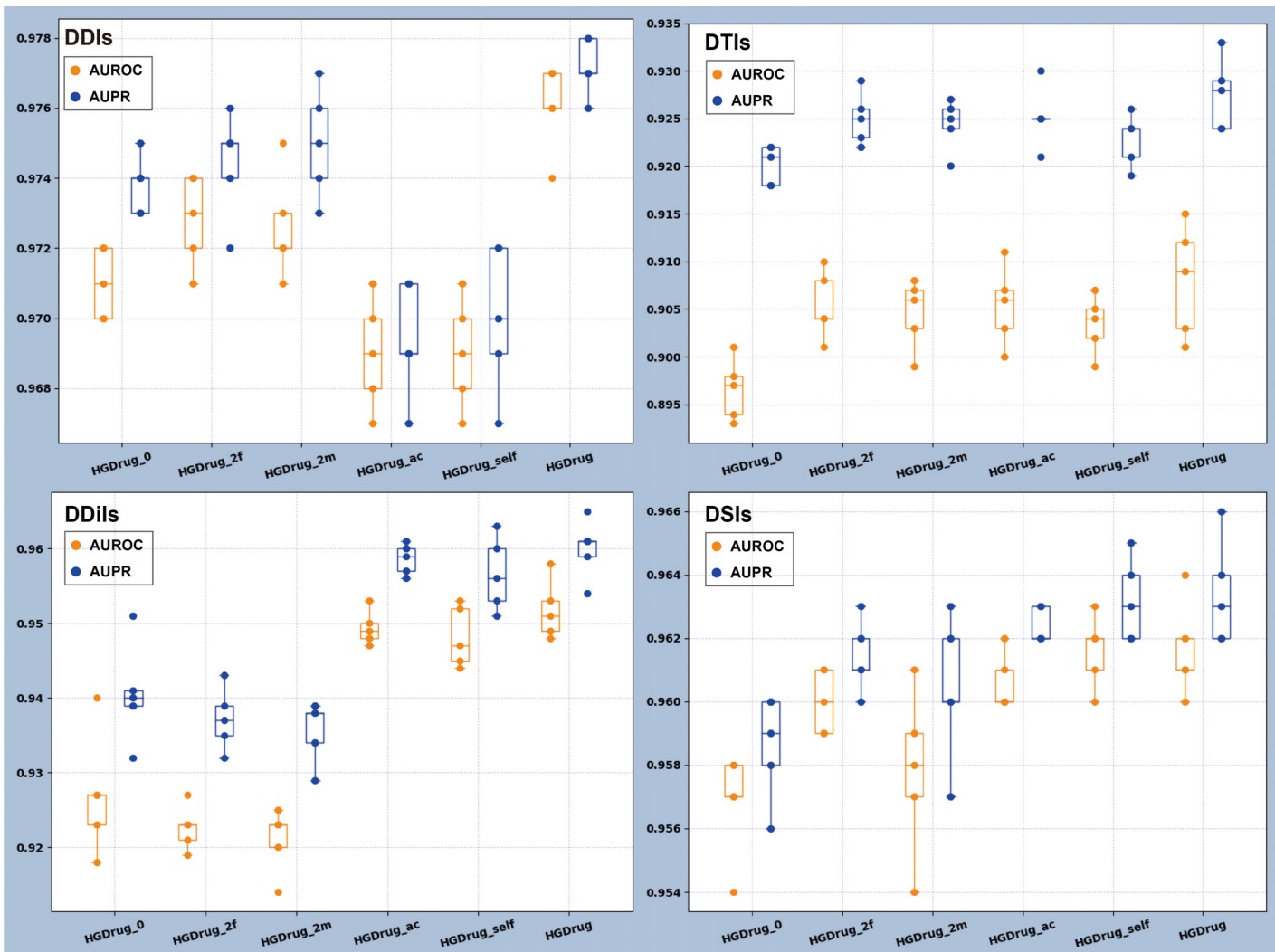

**Fig 4. Ablation experiments explore the contribution of neural network framework and different category hypergraph branches.**

**Table 3. Top 20 novel DDIs predictions and their validation.** If the DrugBank drug interaction records the interaction information between the two drugs, the "label" is set as 1, otherwise 0. The Table E in S1 Text describes the details of DDIs.

| Rank | Drug 1 | Drug 2 | Label | Rank | Drug 1 | Drug 2 | Label |
|---|---|---|---|---|---|---|---|
| 1 | Magnesium salicylate | Liothyronine | 1 | 11 | Picosulfuric acid | Mycophenolate mofetil | 1 |
| 2 | Aliskiren | Dipyridamole | 1 | 12 | Ceritinib | Fosaprepitant | 1 |
| 3 | Silodosin | Diltiazem | 1 | 13 | Dabrafenib | Budesonide | 1 |
| 4 | Calcium carbonate | Levothyroxine | 1 | 14 | Canagliflozin | Sitagliptin | 1 |
| 5 | Netupitant | Zopiclone | 1 | 15 | Metamizole | Adalimumab | 0 |
| 6 | Calcium carbonate | Alfacalcidol | 1 | 16 | Simeprevir | Propafenone | 1 |
| 7 | Acenocoumarol | Ciprofloxacin | 1 | 17 | Lithium cation | Amiloride | 0 |
| 8 | Dabrafenib | Propafenone | 1 | 18 | Dasatinib | Bortezomib | 1 |
| 9 | Dabrafenib | Domperidone | 1 | 19 | Dabrafenib | Bromocriptine | 1 |
| 10 | Dasatinib | Escitalopram | 1 | 20 | Fosphenytoin | Chloramphenicol | 1 |

identify potential drug-side-effects interactions. HAN-DDI [23] is a heterogeneous graph attention model consisting of an attention-based heterogeneous graph node encoder for obtaining drug node representations and a decoder for DDIs prediction. In addition, drug-target interactions (DTIs) and drug-disease interactions (DDiIs) prediction are becoming more and more important for researchers as an integral part of drug repurposing. DTINet [24] learns features of drugs and targets from heterogeneous networks, and then adopts inductive matrix completion to predict novel DTIs and repurpose existing drugs. NeoDTI [25] integrates diverse information from the heterogeneous network and automatically learns representations of drugs and targets for DTIs prediction. Zeng et al. [9] integrate large biomedical network datasets and employ positive-unmarked matrix completion to predict previously unknown DTIs. Prior to that, the researchers constructed a heterogeneous network by integrating 10 networks and learning high-level features of drugs to predict drug-disease interactions [8]. Li et al. [26] propose inductive matrix completion with heterogeneous graph attention network for the prediction of DTIs. Zhao et al. [15] integrate the heterogeneous networks information from the topological and biological perspectives to predict unknown DDiIs. Fu et al. [27] develop a novel link prediction model, multi-view graph convolutional network (MVGCN) for link prediction in biomedical bipartite networks.

These methods, which are based upon heterogeneous networks, have utilized most available network information and learned about the characteristics of the nodes by the associated information on the network topology, in order to identify possible relations between the nodes. However, to the best of our knowledge, most existing methods do not take drug sub-structure information into account and are only effective for a single task.

## Hypergraph

A hypergraph is a generalization of a graph, where an edge can connect any number of vertices and has been widely used in the various fields due to its ability to capture higher-order correlations between nodes in ordinary network data [28–32]. In the field of bioinformatics, hypergraph has also begun to attract more research interest. HypergraphSynergy [33] is a multi-way relation-enhanced hypergraph representation learning method to predict anti-cancer drug synergy. EMPHCN [34] is a drug repositioning method, which is based on enhanced message passing and hypergraph convolutional networks to predict DDiIs. WHGMF [35] uses the generalized matrix factorization based on a weighted hypergraph learning model for microbial-drug association predictions. HHDTI [36] is a heterogeneous hypergraph-based framework

and fuses the key embeddings from a generative model and side embeddings from hypergraph neural networks for drug-target interaction (HHDTI) predictions. Saifuddin et al. [37] decompose the SMILES of drugs to construct a hyperedge and propose the HyGNN model. In their method, different decomposition methods of a drug are regarded as hyperedges and drug representations are learned from different hyperedges. It shares the same concept as learning the representation of a drug from different perspectives to predict DDI, which indicates the potential effectiveness of incorporating drug substructure networks. Still, it does not explore the full characteristics of the hypergraph.

## Preliminary and problem formulation

In this section, we present preliminary concepts and give a formal definition of our target problem.

**Definition 1** (Heterogeneous Network) *The heterogeneous network is denoted $G = (V, E)$ associated with a node type mapping function $\Phi : V \rightarrow O$, where O represents the set of all node types, and an edge type mapping function $\Psi : E \rightarrow R$, where R represents the set of all edge types. Each node $v \in V$ belongs to a particular node type, and each edge $e \in E$ belongs to a particular edge type. If $|O| + |R| > 2$, the network is called a heterogeneous network.*

This paper studies three heterogeneous networks (DTIs, DDiIs, DSIs), and one homogeneous network (DDIs). We integrate the 4 aforementioned networks with drug substructure networks (DFIs, FFIs) obtained from decomposing the drug, to construct the comprehensive drug-centric heterogeneous network DSMN.

**Definition 2** (Network Motifs) *A motif represents a small pattern of interconnections that occurs in complex networks at a frequency significantly higher than in randomized networks, thereby characterizing higher-order network structures [38, 39]. A motif M is defined on n nodes by a tuple $(T, A)$, where T is a $n \times n$ binary matrix and $A \subset \{1, 2, \ldots, n\}$ is a set of anchor nodes.*

In this paper, we construct motifs related to formulate the high-order relation information centered on the drug, which capture the features of the drug in the heterogeneous network. Fig 2(a) shows all the used motifs.

**Definition 3** (Hypergraph) *A hypergraph can be denoted as $\mathcal{G} = (\mathcal{V}, \mathcal{E})$, the vertex set is $\mathcal{V}$ including N unique nodes and the edge set is $\mathcal{E}$ including H hyperedges. Each hyperedge $\epsilon \in \mathcal{E}$ can connect multiple nodes, which means each hyperedge can be represented as a subset of nodes $\epsilon = \{\varsigma_1, \varsigma_2, \cdots, \varsigma_k\}, \varsigma \in \mathcal{V}$, k is the size of hyperedge $\epsilon$. The hypergraph can be represented by an incidence matrix $\mathbf{H} \in \{0, 1\}^{N \times H}$, where $\mathbf{H}(i, j) = 1$ if node $\varsigma_i \in \mathcal{V}$ is in hyperedge $\epsilon_j \in \mathcal{E}$. Let $\mathbf{D}_\varsigma$ and $\mathbf{D}_\epsilon$ be two diagonal matrices, which denote the degrees of nodes and hyperedges, respectively. The $\mathbf{D}_\varsigma(i, i) = \sum_{j=1}^{H} \mathbf{U}_\epsilon(j, j) \cdot \mathbf{H}(i, j)$ is the degree of node $\varsigma_i$, where $\mathbf{U}_\epsilon \in \mathbb{R}^{H \times H}$ is a diagonal matrix, and the diagonal entries are hyperedge weights. In this work, all weights are assigned with 1, thus $\mathbf{U}_\epsilon$ becomes an identity matrix. The $\mathbf{D}_\epsilon(j, j) = \sum_{i=1}^{N} \mathbf{H}(i, j)$ is the degree matrix of hyperedge $\epsilon_j$.*

**Problem 1** (Drug-related Interactions Prediction) *Given a drug-centric heterogeneous network G, which is composed of molecular and drug-substructure interactions networks. Consider a task-specific drug-related interactions network $G_t$, containing $N_t$ nodes v of one type of biomedical entity such as drugs, diseases, proteins, or side effects, etc., and $E_t$ edges e representing interactions between drug and entity nodes. We denote the adjacency matrix of $G_t$ as W, where $W(i, j)$ is 1 if nodes $v_i$ and $v_j$ are connected in the network and otherwise 0. Our goal is to learn a mapping function f of the probability of interaction between nodes $v_i$ and $v_j$ based on the heterogeneous network G. We wish to predict whether $v_i$ and $v_j$ have a given type of pair-wise relation, which has not been observed.*

## Result

### Description of DSMN and HGDrug

As shown in Fig 1, we first use the reaction information of BRICS to decompose the drugs' SMILES into fragments and further decompose the fragments to the smallest indivisible fragment, and retain reaction information between these fragments to construct the drug-fragment interactions (DFIs), and fragment-fragment interactions (FFIs) network. Then, we construct the micro-to-macro drug-centric heterogeneous network DSMN by assembling DFIs, FFIs networks, and four molecular interaction networks, namely, drug-drug interactions (DDIs), drug-target interactions (DTIs), drug-disease interactions (DDiIs), and drug-side-effect interactions (DSIs) networks. As shown in Fig 2, we propose a hypergraph learning framework, termed HGDrug, for drug multi-task interaction predictions. The pipeline of HGDrug includes two key steps:(1) hypergraph construction: to formulate the high-order relation information among drugs, we carefully design the triangular and quadrilateral network motifs with underlying semantics, and these motifs are divided into four specific categories according to the latent semantic relationship to guide the construction of hypergraphs over the heterogeneous network (Fig 2(a)). (2) Multi-branches hypergraph attention and graph convolutional networks inferring drug-related interactions: the hypergraph attention network are used to encode different motifs-guided hypergraphs, and the attention mechanism is used to selectively aggregate information from different branch-specific drug embeddings to form the comprehensive drug embeddings. A graph convolution network on the drug-related interaction graph encodes the drug-related nodes information and complements the drug feature information for task-special drug-related interaction prediction (Fig 2(c)). To thoroughly evaluate the prediction capability of HGDrug, we conduct a variety of experiments, including performance comparison of ours with 8 task specific models and 6 general models, the ablation study to examine the contribution of different category branches, the network visualization, case studies and comprehensive analysis to reveal whether HGDrug can better capture the drug substructure features and predict novel interactions on multiple tasks.

### Datasets and experimental setup

In this section, we provide details on molecular interaction datasets, baseline methods, and experimental setup.

**Datasets.**  We construct the drug-related heterogeneous network by assembling 4 commonly-available drug-related interaction networks which are: (a) drug-drug interactions (DDIs) network; (b)drug-target interactions (DTIs) network; (c) drug-disease interactions (DDiIs) network; (d) drug-side-effect interactions (DSIs) network. In the work, each drug is converted to DrugBank ID from the DrugBank database (v4.3) [40]. In addition, for each drug in these datasets, we use the reaction information of BRICS [2] to decompose the drugs' SIMLES into fragments that contain functional groups, and further decompose these fragments into smaller fragments until they were indivisible. We construct the drug-fragment interactions (DFIs) and fragment-fragment interactions (FFIs) networks with fragments as nodes in the network. One example is illustrated in Fig 1 showing one drug and the corresponding heterogeneous networks. For the specific task of drug-related interaction prediction, we expand the specific task network by decomposing the drug of the network and construct novel two drug substructure networks, namely, drug-fragment interactions (DFIs) network and fragment-fragment interactions (FFIs) network, to perform the associated prediction of downstream tasks.

*DDIs network*. We collect the drug-drug interactions from the BIOSNAP dataset [41] and collect a total of 48,514 interactions connecting 1,514 drugs. We use the reaction information of BRICS to decompose the SMILES string of these drugs into fragments that contain functional groups, and continue to decompose these fragments into indivisible fragments containing functional groups. In the end, 15,016 fragments, 7,974 DFIs, and 83,811 FFIs are retained.

*DTIs network*. We collect the drug-target interactions from the BIOSNAP dataset [41]. The interaction network contains information on which genes (i.e., proteins encoded by genes) are targeted by drugs available on the U.S. market. We collect a total of 15,138 interactions connecting 5,017 drugs and 2,325 targets. Using the same decomposing strategy, finally 46,855 fragments, 28,453 DFIs, and 816,918 FFIs are retained.

*DDiIs network*. We use the drug-disease network constructed by previous research [8]. The network assembles clinically reported or experimentally validated DDiIs network by assembling data from DrugBank [40] and repoDB [42]. We colelct a total of 6,677 interactions connecting 1,519 drugs and 1,229 targets. Using the same decomposing strategy, finally 12,982 fragments, 7,509 DFIs, and 70,574 FFIs are retained.

*DSIs network*. The known drug-induced side-effects are acquired from SIDER [43], which contains information on marketed medicines and their recorded adverse drug reactions. All side effect items are annotated by UMLS [44] vocabularies and converted to Concept Unique Identifier (CUI). In the end, 153,663 DSIs connecting 1,223 drugs with 5,734 side effects are collected in this study, Using the same decomposing strategy, finally 11,135 fragments, 6,330 DFIs, and 58,083 FFIs are retained.

The statistics of the datasets are described in Table 1.

**Implementation details.** We implement HGDrug by the TensorFlow deep learning framework (The code and data of HGDrug are available at: https://github.com/stjin-XMU/ HGDrug). We illustrate the results of each experiment with 5-fold cross-validation and the following metrics are applied to evaluate the performance: AUROC and AUPR. Note that all baseline comparison models are configured to their default setting or best parameter values reported in the previous study. The neural network model was developed and trained utilizing the TensorFlow framework, specifically version 2.2.0. The trainable parameters of our model consist of three parts: drug and drug-related node embeddings, gate parameters, and attention parameters. Regarding drug and drug-related node embeddings, the HGDrug only need to learn the 0-th layer drug embeddings $\mathcal{D}_0 \in \mathbb{R}^{b \times c}$ and drug-related node embeddings $\mathcal{M}_0 \in \mathbb{R}^{t \times c}$. In the HGDrug architecture, a total of nine gates are employed. Each of the gate has parameters of $(c + 1) \times c$, while the attention parameters are of the same size. For the definition of formula symbols, see Methods. During our model training, the dimension of latent factors (embeddings) is empirically set to 100, the depth of the neural network layer is set to 2, the regularization coefficient $\sigma$ is set to 0.01, the hyper-parameter $\lambda$ is set to 0.001, and the batch size is set to 2000. We use the adaptive learning rate to train HGDrug with a initial learning rate 0.001, and all experiments are run up to 200 epochs. The HGDrug framework is trained on $4 \times 2080$Ti GPUs with 32 Intel(R) Xeon(R) CPU E5–2620 v4 @ 2.10GHz on Ubuntu 18.04 platform.

## Baseline comparison models

We evaluate HGDrug on 4 distinct types of drug-related interactions, including DDIs, DTIs, DDiIs, and DSIs. To explore the generality of the HGDrug model and its excellent performance for each drug-related interactions tasks, we select 14 models of 2 types as baseline comparison models, which are 8 task specific models and 6 general models.

To select 8 task specific models for 4 tasks, we pick up 2 up-to-the-date graph neural network models for each task, as illustrated in following:

**DDI predictions task.** **MIRCLE** [45] treats a DDIs network as a multi-view graph and utilizes the graph contrastive learning to capture inter-view molecule structure and intra-view interactions between molecules simultaneous for DDIs prediction; **HyGNN** [37] is a hypergraph neural network model, which predicts the DDIs by generating and using the representation of hyperedges as drugs.

**DTI predictions task.** **IMCHGAN** [26] is an inductive matrix completion with heterogeneous graph attention network, which adopts a two-level neural attention mechanism to learn drug and target feature representations and uses the inductive matrix completion to predict DTIs score; **HHDTI** [36] captures key embeddings and side embeddings via generative model and hypergraph neural networks respectively, and fuses these embeddings to predict DTIs.

**DDiI predictions task.** **DeepDR** [8] constructs the heterogeneous network by integrating 10 networks and learns high-level features of drugs from the heterogeneous networks by a multi-modal deep auto-encoder to predict drug-disease interactions; **HINGRL** [15] integrates the information of heterogeneous biological networks and the biological knowledge of drugs and diseases to obtain node features, and adopts a random forest classifier to predict unknown DDiIs.

**DSI predictions task.** Timilsina et al. [46] integrate the bipartite graph and the semantic similarity graph using a matrix factorization method and a diffusion-based model to discover the DSIs. In this work, we use **MFD** to refer to this model; **MGPred** [21] uses graph attention network to integrate three different types of features, which includes similarity information, known DSIs information, and word embedding for DSIs prediction.

The 6 general models suitable for all tasks including the classic machine learning algorithm **SVM** [47], **Katz** [48], the direct network embedding methods **Deepwalk** [49], the deep graph neural network model **GCN** [50] and **GAT** [51], and the latest model **SkipGNN** [52] for molecular interaction association prediction. For these models details see Details about "General models baselines" in S1 Text.

## Performance of HGDrug on the cross-validation

We firstly select the task-specific model for each task for comparison, including comprehensive network models, path-based models, hypergraph models, and considering drug substructures models. For models that can expand network data, we add the DFIs network data as a variant of the original model to verify whether or not the chemical structure information can effectively improve the performance for drug-related interactions prediction. We discover that HGDrug outperforms each of these competitive baseline models, by consistently achieving the highest prediction accuracy for all 4 tasks. HyGNN model [37] is not able to capture high-level information between drugs since it simply uses the different decomposition schemes of each drug as a hyperedge, and constructs a hypergraph of each drug to learn the feature of the drug. It is worth noting that the performance of the original network-based model will be improved by incorporating the dataset from DFIs network, especially for the path-based model. By treating drug substructures that contain pharmacophore information as node types and integrating them into heterogeneous networks, we can enhance the GNN model's ability to discern relationships between drugs with potential similar efficacy, thereby improving the prediction performance in drug-related tasks. For example, the IMCHGAN [26] model, which introduces meta-path-based neighbors to learn connections between nodes from different semantic perspectives, yields a performance improvement of increasing AUROC from 0.800 to 0.890, and AUPR from 0.796 to 0.902 on the DTIs dataset respectively when the DFIs network is added.

These experiments also show that for drug-related interaction prediction tasks, adding DFIs network can learn information about the same functional group fragments between drugs, which helps to improve the performance of drug-related interaction prediction based on network models. Since the HGDrug makes full use of the higher-order relations between nodes in heterogeneous networks and the information of drug substructures, it outperforms other state-of-the-art methods in drug-related interaction prediction performance, even if these methods add substructure networks.

In addition, we compare HGDrug's performance with 6 general baseline methods. The experimental results are reported in Table 2. It can be seen that HGDrug is the top-performing method out of 6 general methods across all drug-related interaction networks. In the DDIs and DDiIs tasks, HGDrug has achieved a significant performance improvement than the baseline methods. Compared to the second-ranked baseline method, the AUROC and AUPR of the DDIs task are improved by 11.8% and 14.3%, respectively, and the AUROC and AUPR of the DDiIs task are improved by 11.3% and 9.9%, respectively. Note that mining potential drug-drug associations for the DDIs task might be helpful since drug substructure networks brings extra complexities to the network. Drugs with the same basic structure are more likely to have similar pharmacological effects, which indicates that incorporating drug substructure networks can help identify new drug indications. Compared to classical machine learning and the direct embedding methods, the AUROC of HGDrug has up to 36.3% increase over SVM, 31.1% increase over Katz, and 25.4% increase over DeepWalk on DDIs task. These results also reflect that deep graph neural networks can learn to capture more effective features than general machine learning methods and direct embedding methods. It is interesting to observe that the well-performing benchmark, SkipGNN, has worse performance than GCN on the DTIs and DDiIs tasks, which might be due to the fact that DTIs and DDiI have fewer known interactions, while skipGNN skips similarity thus losing its performance advantage. For more evaluation metrics results, see "More evaluation metrics" in S1 Text.

## Performance of HGDrug by ablation analysis

We implement 5 simplified variants of HGDrug to exam the effects of each component in our HGDrug in the 4 distinct types of prediction tasks. We run all the experiments with the 5-fold cross-validation, same model parameters and evaluation protocol (see the Implementation details) for a fair comparison. The result is presented in Fig 4. The 5 simplified variants of HGDrug are denoted as:

- HGDrug_0: removing the $1 \sim 4$-th all hypergraph branches.

- HGDrug_2f: removing the 3-th and 4-th molecular interactions hypergraph branches and retaining the 1-th and 2-th drug-substructure hypergraph branches.

- HGDrug_2m: removing the 1-th and 2-th drug-substructure hypergraph branches and retaining the 3-th and 4-th molecular interactions hypergraph branches.

- HGDrug_ac: obtaining the drug representation by hypergraph convolutional network.

- HGDrug_self: removing the self-supervised auxiliary task.

Fig 4 shows only minor difference obtained by using drug-substructure hypergraph branches or molecular interactions hypergraph branches alone, which proves that the drug information captured by the drug-substructure networks can rival the drug feature information obtained from the drug-related interaction network. For DDiI tasks with fewer known interactions, a large difference can be observed whether all hypergraph branches are utilized to

obtain features (HGDrug_ac, HGDrug_self, HGDrug) or not (HGDrug_0, HGDrug_2m, HGDrug_2f). It illustrates the importance of obtaining more reliable side information when there are insufficient known interactions for the association prediction of molecular interactions. HGDrug uses the hypergraph attention network architecture to focus on the differences of neighbor features in the hypergraph, and uses a self-supervised auxiliary task to weigh the contribution of different types of high-order information in interaction predictions. Therefore, HGDrug can learn higher-quality drug features information from the hypergraph, which is superior to the HGDrug_ac that obtains drug representation through the hypergraph convolutional network and HGDrug_self that removes the self-supervised auxiliary task. These studies also show that additional feature information can be captured by the proposed HGDrug to enhance drug-related interaction predictions, which may provide new insights into understanding interaction mechanisms among drugs, drug substructure, and molecules. To more thoroughly evaluate the significance of utilizing fragments as a basis for hypergraph construction in drug feature learning, we verify the contribution of the four motifs ($M_2$, $M_3$, $M_5$, $M_6$), the results see Table C in S1 Text.

## HGDrug identifies novel associations

The main goal of the HGDrug model is to discover potential novel interactions based on existing known interactions. To further demonstrate the ability of HGDrug for discovering novel interactions, we select the prediction results of DDiIs and DDI tasks for analysis.

**Network visualization of the DDiI predictions.**   We remove the known DDiIs used in the prediction model and retain the novel top 100 DDiIs with the highest drug-disease association score predicted by HGDrug (Fig 5(a)). Among the top 100 predicting DDiIs, HGDrug can capture the experimental or clinically reported drug-disease associations. For example, the five results with the highest association score predict that drug DB01413 (Cefepime) can act on indications C0178299 (Infection of skin AND/OR subcutaneous tissue) and C0276026 (Haemophilus influenza pneumonia), DB00535 (Cefdinir) can also act on C0178299, DB00833 (Cefaclor) can act on C0554628 (Streptococcus pyogenes infection), and DB01331 (Cefoxitin) can act on C0149725 (Lower respiratory tract infection). As a widely known fact, Cefepime is a fourth-generation cephalosporin antibiotic, which can treat uncomplicated skin and skin structure infections [53–55] and pneumonia caused by susceptible bacteria [56]; Cefdinir is a third-generation cephalosporin, which can also be used to treat uncomplicated skin and skin structure infections [57]; Cefaclor is a second-generation cephalosporin that can be used to treat streptococcus pyogenic infection [58]; Cefoxitin is a semi-synthetic, broad-spectrum antibiotic that can be used to treat lower respiratory tract infection [59]. The above results prove the effectiveness of our model in finding novel potential associations. If properly utilized, it can assist medicinal chemistry learning to discover new associations of approved drugs.

HGDrug predicts novel interactions by learning more effective drug features. To verify the hypothesis and interpret the results of our HGDrug model, we analyze the above 5 novel associations from the perspective of drug features learned by HGDrug. We extract the drug features obtained in the DDiIs task and calculate the similarity scores between drugs. The result shows that the most similar drugs for 4 drugs DB01413 (Cefepime), DB00535 (Cefdinir), DB00833 (Cefaclor), and DB01331 (Cefoxitin) are DB01330 (Cefotetan), DB00833 (Cefaclor), DB01416 (Cefpodoxime), and DB01330 (Cefotetan), respectively. We give the top 20 highest similarity score DDIs of these 4 drugs (see Table D in S1 Text). We discover that these 4 drugs with the most similiar drug all are the approved drugs for the predicted corresponding indications (see Fig 5(b)). These results demonstrate that our model can make full use of features

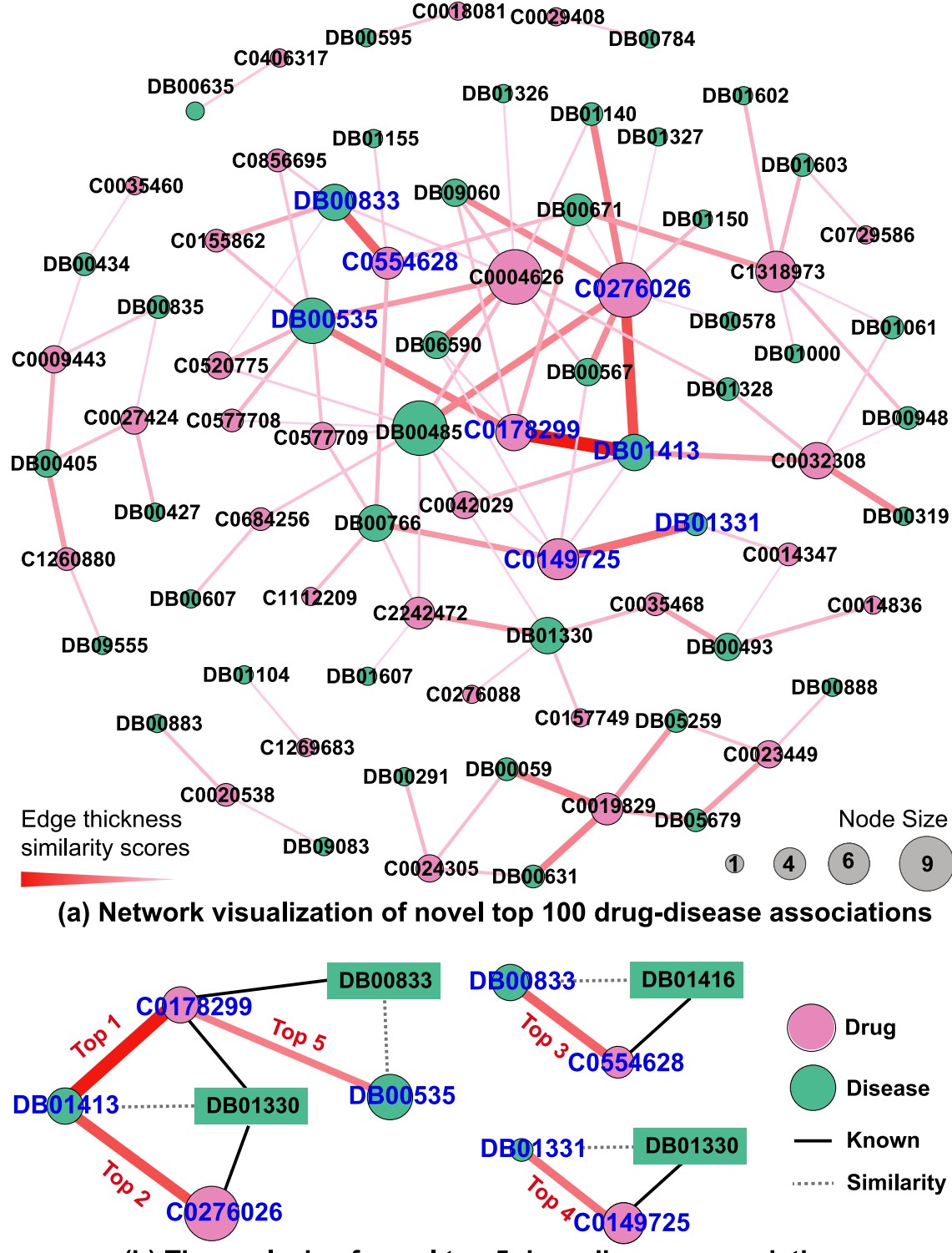

**(a) Network visualization of novel top 100 drug-disease associations**

**(b) The analysis of novel top 5 drug-disease associations**

**Fig 5. Network visualization and analysis of the drug-disease associations predicted by HGDrug.** (a) In the network, the predicted novel top 100 DDiI network is visualized. The pink nodes are diseases and the green nodes are drugs. The label of the node represents the ID of the drugs (Drugbank_ID) or diseases (UMLS_ID). The node size denotes the degree. The weight of edges (drug-disease associations) denotes the predicted score by HGDrug. (b) Analysis of top 5 drug-disease associations predicted by HGDrug. Four drugs are included in the top 5 predicted potential associations. The similarity between drugs comes from the drug features learned by

HGDrug in the DDiI task, and the drug most similar to these four drugs are selected respectively. This network was generated by Gephi (https://gephi.org).

information to capture approximate relations between drugs, and reveal potential drug-disease associations based on the relations and the approved drug-disease interactions.

The chemical structures are responsible for specific properties of drugs, and the same basic chemical structure exist in drugs with the same pharmacological effect. Accordingly, identifying the drug substructure information associated with the target property and adding the information into the model is vital for drug discovery. To illustrate HGDrug's learning ability of chemical structures, we visualize the above 4 pairs of drug with approximate relations. It's worth noting that these drug pairs share the same substructures that carry pharmacophore information, see Fig 6 for more details. This experimental result also shows that HGDrug can learn the substructure information and capture the same basic chemical structure information to improve the performance of drug repurposing.

**Case study of the DDI predictions.** In this part, we remove the known DDIs used in the prediction model and retain the top 20 novel interactions predicted in the DDIs task. Table 3 displays the top 20 novel predicted DDIs and information showing association between two drugs recorded in the DrugBank database (https://go.drugbank.com). It can be seen from Table 3 that among the top 20 novel predicted DDIs, 18 can be found in the DrugBank's drug interactions records. The result indicates the HGDrug model can effectively identify potential novel interactions. To intuitively observe the drug representations that our hypergraph branches have learned, we envision the representations by mapping them to the two-dimensional space by t-SNE algorithms. The result see Fig A in S1 Text, the implementation details see "Visualization details" in S1 Text. Remarkably, even without any label information, the

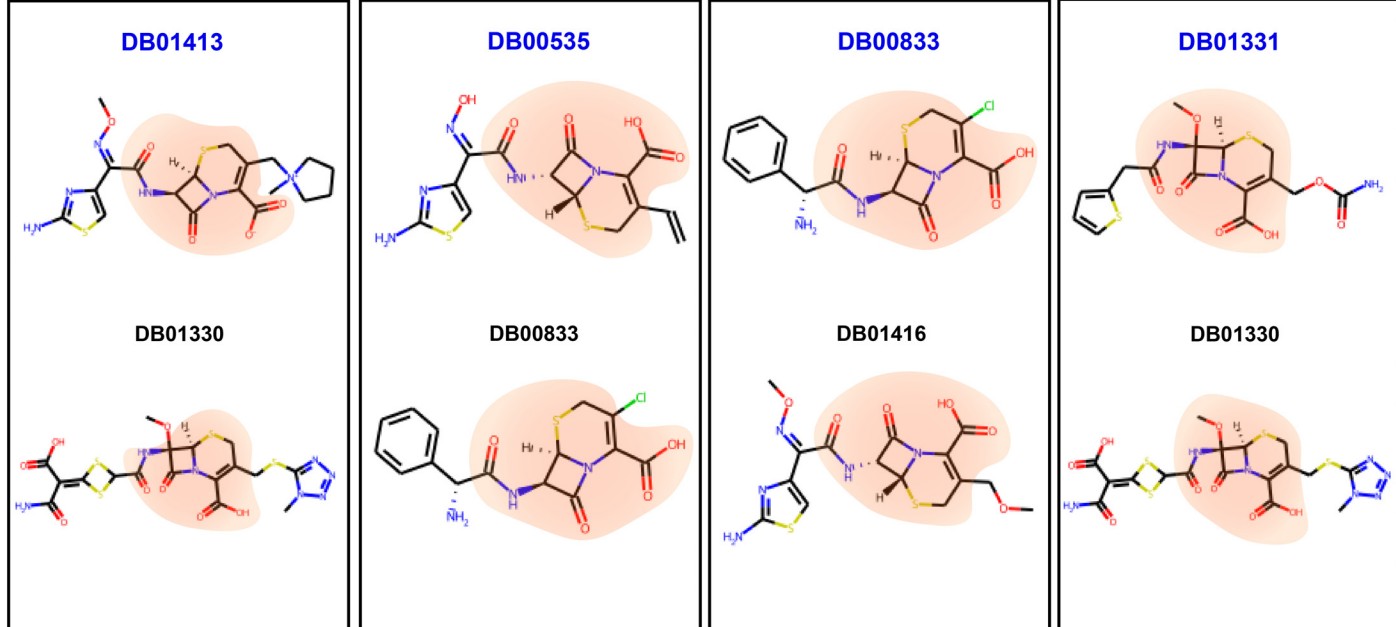

**Fig 6. The 4 drugs and their corresponding top 1 drug structures from Fig 5(b).** The same substructures of drug pairs are marked in orange.

representations learned from hypergraph branches follow some kind of pattern that is strongly related to the drug classes.

Furthermore, we explore the ability of different category hypergraph branches to identify potential DDIs. We select the top 20 novel interactions based on the predicted highest association from the HGDrug_2f and HGDrug_2m. It can be seen that among the top 20 novel interactions of the HGDrug_2f, 19 can be found in the DrugBank's drug interactions records (see Table F in S1 Text). And 15 of the top 20 novel interactions in the HGDrug_2m model can be found in DrugBank's drug interactions records (see Table G in S1 Text). The experimental result thus illustrates the importance of drug substructure in identifying potential DDIs.

## Comprehensive analysis: Computationally identified the most relevant entities of paclitaxel

To further illustrate the practical applications of HGDrug, we comprehensively analyze the prediction potential of HGDrug from the perspective of drug, target, side effect and disease. As a widely used and promising anticancer drug, paclitaxel is a type of microtubule stabilizer and used to treat various diseases, including cancers (e.g. breast cancer, ovarian cancer), coronary artery restenosis, renal and hepatic fibrosis [60]. However, severe side effects limit the effectiveness of paclitaxel in the disease treatment, such as myelosuppression and peripheral neuropathy [61]. Therefore, we select paclitaxel as a case for further comprehensive analysis, and predict its potential drugs, targets, side effects, and diseases using HGDrug. For the results of the DTIs prediction task, we do not consider the CYP series of proteins.

As shown in Fig 7, the most relevant entities predicted by paclitaxel in the four tasks are P00372 (ESR1), multiple myeloma, dysaesthesia, and cobicistat. The ESR1, multiple myeloma, and dysaesthesia have almost no drugs shared with paclitaxel, which show that these results cannot be effectively predicted by GNNs that relying on the pairwise links. Recent studies have shown that ESR1 methylation may have a protective effect on neurotoxicity induced by paclitaxel [62]. Paclitaxel have shown therapeutic effect on advanced breast cancer patients with ESR1 mutations [63], which indicate HGDrug provided insight into the potential mechanism for known drug-disease (side effect) interactions. In addations, we analyse the top 10 potential targets and find that 7 targets indeed associate with paclitaxel validated by literature (see the Table H in S1 Text). Several studies have shown that paclitaxel induces dysaesthesia, including pain, allodynia, and numbness [64, 65]. Furthermore, Table I in S1 Text provides the top 10 predicted side effects, and find most side effects are validated to relate to paclitaxel by the literature. It is generally recognized that the identification of drug-disease interactions is laborious and expensive. Several studies reported that paclitaxel selectively modifies the expression of regulatory proteins in the apoptosis signal pathway and can be used to treat multiple myeloma [66, 67]. Furthermore, Bcl-2 is proved to associate with resistance to paclitaxel in multiple myeloma cells, which may relate to its influence on paclitaxel-induced apoptosis [68, 69]. These results indicate that HGDrug has the potential to discover new drug-disease interactions from the perspective of mechanism. To further prove the role of hypergraph and substructure, we give the substructure of the most relevant cobicistat and paclitaxel, and we can find that they have the same substructure information, and this information can be captured through the hypergraph structure with "DISS" semantics.

## Discussion and conclusion

The biomedical networks which contain domain knowledge and can be used for the prediction of various tasks such as drug repurposing and drug-drug interaction, however, chemical that play important role in drug properties is neglected in current biomedical networks.

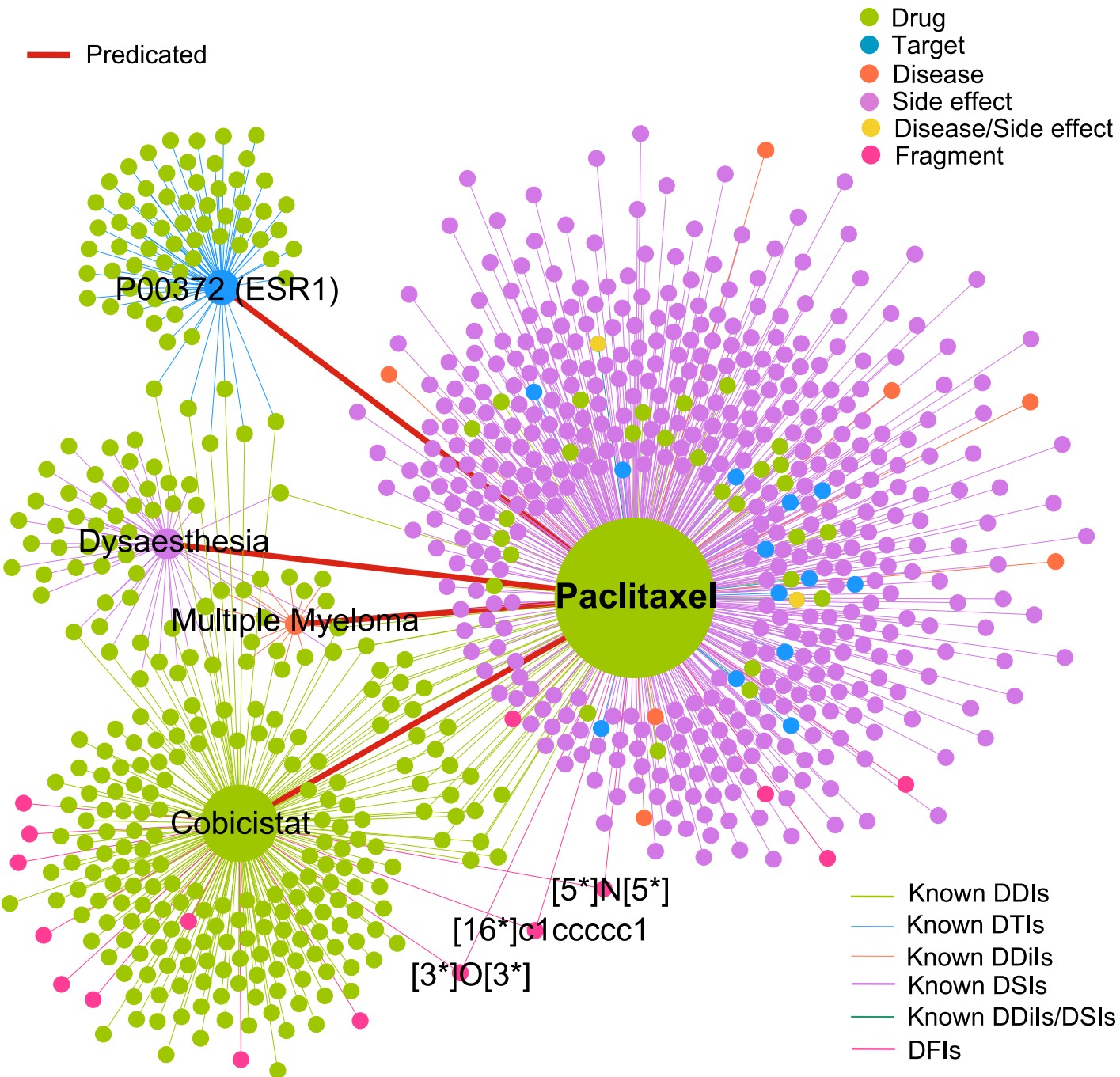

**Fig 7. Analysis of the predicted results for the four tasks of paclitaxel.** The network contains the known interactions of paclitaxel and the most relevant entity nodes predicted by HGDrug for paclitaxel in the four tasks. The known related drugs of four nodes are given for visualizing the drugs that these nodes share with paclitaxel.

Furthermore, previous drug-related interaction prediction methods mostly rely on pairwise links heavily, without paying attention to local strong connections in complex biological networks, and cannot correlation between drugs and chemical structures. In the work, we propose the micro-to-macro drug-centric heterogeneous network method, and develop a general framework, called HGDrug, which is a novel hypergraph attention network for predicting

drug-related interactions. Firstly, we construct a novel drug-centric network called DSMN that incorporates information on drug substructures into common drug interaction data. We then design 12 types of network motifs and divide these motifs into 4 categories based on whether the drugs are directly related, and whether they are derived from substructure or molecular interaction networks, based on which 4 motifs-driven hypergraphs are constructed. HGDrug model applies multi-branches hypergraph attention to learn high-order drug features representation from motifs-driven hypergraphs, and uses graph convolution network on the known drug-related interaction graph to obtain the drug-related nodes information and to complement the drug feature information for task-special drug-related interaction prediction. The experimental results show that the data supplementation of drug substructure and the multi-branch hypergraph attention mechanism can better obtain the characteristic results of drugs and improve the prediction performance of drug-related interaction networks. By comparing our work with multi-task general models and specific task models, it can be seen that HGDrug achieves excellent and robust performance on various prediction tasks of drug-related interaction networks. Predictive analytics for novel interactions of DDiIs and DDIs further prove the effectiveness of HGDrug in finding novel drug-related interactions.

While we have explored the interpretability of the predicted results, not all chemical structures that embody the physicochemical properties of the drug can be found in all cases. As previously mentioned, pharmacophore information contained in chemical structures is crucial for effective drug discovery, and a more comprehensive investigation from this aspect is planned in our future studies. In addition, fragment-based drug discovery (FBDD) has become one of the mainstream methods for discovering lead compounds in recent years. Further exploration of network construction method is also necessary for fragment-based drug discovery, which may help GNN models discover potential molecular fragments for specific diseases and targets.

In summary, our model can be used as an effective method to predict drug-related interactions, to develop a new idea for drug-related interactions calculation, and to provide computer-aided guidance for biologists in clinical trials. Furthermore, the hypergraph-based method is not constrained by the size of the network data. It leverages existing network data to extract strong correlations between drugs, thereby enhancing the predictive accuracy in drug-related tasks. This approach allows for the expansion of calculations to larger network datasets without escalating computational complexity. Moreover, HGDrug is adept at handling most drug-related interaction task predictions, significantly minimizing the consumption of computing resources and facilitating a more convenient and efficient execution of related tasks.

## Methods

### Motifs-driven hypergraph

We decompose the drugs and construct the interaction network between drugs and substructures in order to mine the feature information of drugs and capture potential associations between drugs through the number of drugs with the same substructure and the inclusion relationship between substructures.

To capture higher-order information among drugs, we first align these heterogeneous networks and molecular interaction network for specific prediction task and constructs hypergraphs based on the network motifs over heterogeneous networks (Fig 2(a)). In the work, the DFIs and FFIs networks obtained by decomposing the drugs are directed, the drug-related heterogeneous networks are directed, and the DDIs network is undirected (bidirectional). We focus on triangle and quadrilateral motifs representing high-order relationships between drugs and carefully designed a set of motifs to guide the hypergraph construction. The $M_1$-$M_6$

summarizes the common triangle and quadrilateral relations in the DDIs, DFIs, and FFIs networks, describing the high-order correlation for drugs that have the same substructure. In addition to drug substructure networks, we also build hypergraphs based on molecular interaction networks, focusing only on triangular motifs in these heterogeneous networks. The $M_7$-$M_{12}$ summarizes the common triangles in the molecular interaction networks. Last while not least, we also take whether the drugs are directly related into consideration, and subdivide the above 2 groups into following groups: $M_1$-$M_3$ (drug-related and have the same substructure, DRSS), $M_4$-$M_6$ (drug independent and have the same substructure, DISS), $M_7$-$M_9$ (drug-related and have the same molecular interactions, DRSM), $M_{10}$-$M_{12}$ (drug independent and have the same molecular interactions, DISM). Fig 2(a) shows all the used motifs, it is worth noting that for better personal data protection, we only use the task specific training dataset to build its associated hypergraph in a specific task.

For a motif $M$, we use the co-occurrence of two nodes in $M$ to capture the higher-order relations of drugs. Specifically, given a motif $M_k$ where $\mathbf{A}_k$ represents the motif-based adjacency matrix, $\mathbf{A}_k(v_i, v_j)$ refers to the number of times that $v_i$ and $v_j$ appear in the same motif structure of $M_k$ in the global heterogeneous network. The motif-based adjacency matrix for node type $O_i$ in the heterogeneous is denoted as:

$$\mathbf{A}_k(v_i, v_j) = \sum_{v_i, v_j \in V} \mathbf{1}(v_i, v_j \text{ occur in } M_k) \tag{1}$$

where $i \neq j$, the $\Phi(v_i) = \Phi(v_j) = O_i$ is considered as the drug node type, and $\mathbf{1}()$ is the truth-value indicator function, in a way that when the statement inside () is true, $\mathbf{1}() = 1$, otherwise 0. The Table 4 shows how the matrix $\mathbf{A}_k$ is calculated with respect to $M_k$ motif. Since drug independent motifs $M_4$-$M_6$ and $M_{10}$-$M_{12}$ are believed to contain the corresponding triangular or quadrilateral drug related motifs $M_1$-$M_3$ and $M_7$-$M_9$, we remove the redundancy and eventually make $\mathbf{A}_4 = \mathbf{A}_4 - \mathbf{A}_1$, $\mathbf{A}_5 = \mathbf{A}_5 - \mathbf{A}_2$, $\mathbf{A}_6 = \mathbf{A}_6 - \mathbf{A}_3$, $\mathbf{A}_{10} = \mathbf{A}_{10} - \mathbf{A}_7$, $\mathbf{A}_{11} = \mathbf{A}_{11} - \mathbf{A}_8$, and $\mathbf{A}_{12} = \mathbf{A}_{12} - \mathbf{A}_9$

**Table 4. Computation of motif-induced adjacency matrices.**

| Motifs $M$ | Matrix Computation $\mathbf{A}$ |
|---|---|
| $M_1$ | $\mathbf{A}_1 = (\mathbf{SS}) \odot \mathbf{S}$ |
| $M_2$ | $\mathbf{A}_2 = (\mathbf{YY}^T) \odot \mathbf{S}$ |
| $M_3$ | $\mathbf{A}_3 = \mathbf{A}_m + \mathbf{A}_m^T, \mathbf{A}_m = (\mathbf{YZY}^T) \odot \mathbf{S}$ |
| $M_4$ | $\mathbf{A}_4 = \mathbf{SS}$ |
| $M_5$ | $\mathbf{A}_5 = \mathbf{YY}^T$ |
| $M_6$ | $\mathbf{A}_6 = \mathbf{A}_m + \mathbf{A}_m^T, \mathbf{A}_m = \mathbf{YZY}^T$ |
| $M_7$ | $\mathbf{A}_7 = \mathbf{WW}^T \odot \mathbf{S}$ |
| $M_8$ | $\mathbf{A}_8 = \mathbf{GG}^T \odot \mathbf{S}$ |
| $M_9$ | $\mathbf{A}_9 = \mathbf{VV}^T \odot \mathbf{S}$ |
| $M_{10}$ | $\mathbf{A}_{10} = \mathbf{WW}^T$ |
| $M_{11}$ | $\mathbf{A}_{11} = \mathbf{GG}^T$ |
| $M_{12}$ | $\mathbf{A}_{12} = \mathbf{VV}^T$ |

$\mathbf{A}$: motif-driven adjacency matrix;

$\mathbf{S}$: DDI adjacency matrix; $\mathbf{Y}$: DFI adjacency matrix;

$\mathbf{Z}$: FFI adjacency matrix; $\mathbf{W}$: DTI adjacency matrix;

$\mathbf{G}$: DSI adjacency matrix; $\mathbf{V}$: DDiI adjacency matrix;

$\odot$: denotes the element-wise product.

With the following 4 groups of motifs, e.g., RSS, ISS, RSM, ISM, we construct four hypergraphs that contain different high-order drug relation patterns. We use the incidence matrices $\mathbf{I}_j = \mathbf{A}_1 + \mathbf{A}_2 + \mathbf{A}_3$, $\mathbf{I}_p = \mathbf{A}_4 + \mathbf{A}_5 + \mathbf{A}_6$, $\mathbf{I}_i = \mathbf{A}_7 + \mathbf{A}_8 + \mathbf{A}_9$, and $\mathbf{I}_u = \mathbf{A}_{10} + \mathbf{A}_{11} + \mathbf{A}_{12}$ to represent 4 hypergraphs induced by the above 4 motif groups, respectively. Take $\mathbf{I}_j$ (based on the heterogeneous network shown in Fig 1) as an example, the related $M_1$-$M_3$ and the final incident matrix $\mathbf{I}_j$ obtained are shown in the (Fig 2(a)).

## Learning drug representation by multi-branches hypergraph attention

Our representation learning method is developed upon MHCN model [32]. Based on the 4 groups of motif-driven hypergraphs obtained in the previous section, 4 branches are used to extract multi-scale drug features of the given heterogeneous network, with each branch capturing high-order drug relationship patterns in a motif-driven hypergraph. To distinguish higher-order drug-drug relations in different hypergraphs to the final drug-related interaction prediction performance, we apply a filtered self-gating unit for pre-processing when drug initial feature vectors matrix $\mathcal{D}_0$ are the input for each branch, as defined in following:

$$\mathcal{D}_0^x = \mathcal{D}_0 \odot \sigma(\mathcal{D}_0 \mathbf{P}_g^x + \mathbf{B}_g^x) \tag{2}$$

where $x \in \{j, p, i, u\}$ represents the branches, $\odot$ is the element-wise product, $\sigma$ is the sigmoid nonlinearity, $\mathbf{P}_g^x \in \mathbb{R}^{c \times c}$ is the parameters matrix to be learned, $c$ is the dimension of the drugs initial feature, and $\mathbf{B}_g^x \in \mathbb{R}^c$ is the parameters vector to be learned.

After the nonlinear gate modulates the drug's initial features at a feature-wise granularity through dimension re-weighting, we then obtain the branch-specific drug features $\mathcal{D}_0^x$. The hypergraph attention on each branch is defined as:

$$\mathcal{D}_{l+1}^x = (\mathbf{D}^x)^{-1} relu(\mathcal{D}_l^{x'} \mathbf{P}_m^x + \mathbf{B}_m^x) \tag{3}$$

where $l$ is the propagating layer, $\mathcal{D}_l^{x'} = softmax(\mathbf{I}_x \mathcal{D}_l^x (\mathcal{D}_l^x)^T) \mathcal{D}_l^x$, $\mathbf{D}^x \in \mathbb{R}^{b \times b}$ is the degree matrix of $\mathbf{I}_x$, $b$ is the number of drugs. Note that $\mathbf{I}_x$ can be replaced with arbitrary $\mathbf{I}_j$, $\mathbf{I}_p$, $\mathbf{I}_i$ and $\mathbf{I}_u$ to learn drug representations that encode higher-order information in the corresponding branch using a hypergraph attention network. $\mathbf{P}_m^x \in \mathbb{R}^{c \times c}$ is the parameters matrix to be learned, $\mathbf{B}_m^x \in \mathbb{R}^c$ is the parameters vector to be learned. After the drug embedding is propagated through the $L_{th}$ layer, the embedding features obtained by each layer are averaged to avoid over-smoothing, and the final representation of the drug obtained in each branch is $\mathcal{D}^x = \frac{1}{L+1} \sum_{l=0}^{L} \mathcal{D}_l^x$.

Moreover, the attention mechanism is applied to aggregate the drug feature information obtained from each branch because the drug features obtained from different branch are not equally important. For each drug $d$ in different branch, a $\omega^x$ is learned to measure the different contributions of branch-specific embeddings to the final prediction performance. The $\omega^x$ is defined as:

$$\omega^x = \frac{exp(\mathbf{a}^T \cdot (\mathcal{D}^x \mathbf{P}_a))}{\sum_{x' \in \{j,p,i,u\}} exp(\mathbf{a}^T \cdot (\mathcal{D}^{x'} \mathbf{P}_a))} \tag{4}$$

where $\mathbf{a} \in \mathbb{R}^c$, $\mathbf{P}_a \in \mathbb{R}^{c \times c}$ are trainable parameters, and the four-scale drug feature are aggregated as $\mathcal{D}^s = \sum_{x \in \{j,p,i,u\}} \omega^x \mathcal{D}^x$.

## Drug multi-task interaction predictions

For the downstream task of drug multi-task interaction prediction, we use graph convolution to obtain the drugs and drug-related nodes information from the known interactions data of

specific downstream tasks. $\mathcal{D}^c$ is defined as the drug feature from the graph convolution, and can be computed as:

$$\begin{cases} \mathcal{D}_0^c = \mathcal{D}_0 \odot \sigma(\mathcal{D}_0 \mathbf{P}_g^c + \mathbf{b}_g^c) \\ \mathcal{D}_{l+1}^c = (\mathbf{D}_d^c)^{-1} relu((\mathbf{R}\mathcal{M}^l)\mathbf{P}_m^c + \mathbf{b}_m^c) \end{cases} \qquad (5)$$

where $\mathcal{D}_l^c$ is the gated drug embedding for graph convolution, $\mathbf{R} \in \{\mathbf{S}, \mathbf{W}, \mathbf{G}, \mathbf{V}\}$ is drug-related interaction adjacency matrix for different tasks, $\mathbf{D}^c \in \mathbb{R}^{b \times b}$ are degree matrices of $\mathbf{R}$, $\mathbf{P}_m^c \in \mathbb{R}^{c \times c}$ is the parameters matrix to be learned, and $\mathbf{b}_m^c \in \mathbb{R}^c$ is the parameters vector to be learned. $\mathcal{M}$ is defined as the drug-related node feature from the graph convolution, and can be computed as:

$$\begin{cases} \mathcal{D}_l^m = \sum_{x \in \{j,p,i,u\}} \omega^x \mathcal{D}_l^x + \dfrac{1}{2}\mathcal{D}_l^c \\ \mathcal{M}_{l+1} = (\mathbf{D}_m^c)^{-1} relu((\mathbf{R}^T \mathcal{D}_l^m)\mathbf{P}_t^c + \mathbf{b}_t^c) \end{cases} \qquad (6)$$

where $\mathcal{D}_l^m$ is the drug embedding, which combines the drug feature information from the above multi-branches hypergraph $\sum_{x \in \{j,p,i,u\}} \omega^x \mathcal{D}_l^x$ and the drug feature information from the graph convolution $\mathcal{D}_l^m$, $\mathbf{D}_m^c \in \mathbb{R}^{t \times t}$ are degree matrices of $\mathbf{R}^T$, and $t$ is the number of drug-related nodes. The $\mathbf{P}_t^c \in \mathbb{R}^{c \times c}$ is the parameters matrix to be learned, and $\mathbf{b}_t^x \in \mathbb{R}^c$ is the parameters vector to be learned. Note that the dimension of the nodes initial feature equals to the dimension of the drug initial feature.

We can therefore obtain the final drug and drug-related node feature embeddings $\mathcal{D}$ and $\mathcal{M}$ by:

$$\begin{cases} \mathcal{D} = \mathcal{D}^s + \dfrac{1}{L+1}\sum_{l=0}^{L} \mathcal{D}_l^m \\ \mathcal{M} = \dfrac{1}{L+1}\sum_{l=0}^{L} \mathcal{M}_l \end{cases} \qquad (7)$$

where $\mathcal{D}_0 = \{\mathbf{d}_1, \mathbf{d}_2, ..., \mathbf{d}_b\}, \mathbf{d} \in \mathbb{R}^c$ and $\mathcal{M}_0 = \{\mathbf{m}_1, \mathbf{m}_2, ..., \mathbf{m}_t\}, \mathbf{m} \in \mathbb{R}^c$ are randomly initialized.

## Objective functions optimizaion

The model applies the Bayesian Personalized Ranking (BPR) loss [70], which is computed as:

$$L_s = \sum_{(d,i,j) \in \Omega} -log\sigma(\hat{s}_{d,i}(\theta) - \hat{s}_{d,j}(\theta)) + \delta \|\theta\|_2^2 \qquad (8)$$

where $\Omega = \{(d, i, j) | (d, i) \in \mathcal{R}^+, (d, j) \in \mathcal{R}^-\}$ denotes the pairwise training data, $\mathcal{R}^+$ indicates the observed interactions, $\mathcal{R}^-$ indicates the unobserved interactions, $\sigma(\cdot)$ is the sigmoid function, $\hat{s}_{d,i} = \mathbf{d}_d^T \mathbf{m}_i, \hat{s}_{d,j} = \mathbf{d}_d^T \mathbf{m}_j$ is the predicted score of $d$ on $i$ and $j$, respectively. $\theta$ is the parameters of model, $\|\theta\|_2$ is the $L_2$ norm of the parameter vector, and $\delta$ is the hyper-parameter that controls the $L_2$ regularization strength to prevent overfitting. The BPR is a pairwise loss that promotes an observed interaction to be ranked higher than its unobserved counterparts. Let drug $d$ be the input to the mode. Assume there is a random positive sample $i$ related to drug $d$, and a random negative sample $j$ unrelated to the drug $d$, the goal of the model optimization is then to yield a higher rank for sample $i$ than sample $j$ in the predicted list of drug $d$.

The different branches can learn drug features with different distributions on different hypergraphs. To avoid the loss of higher-order information possibly caused by aggregation operations, a self-supervised auxiliary task is employed to enhance the performance of the association prediction task. Inspired by the Deep Graph Infomax (DGI) method [71], our model calculates the mutual information between node representations and full graph representations (in large-scale graph networks, sampled sub-graph representations are used). In the self-supervised auxiliary task, a comprehensive drug representation reflecting the local and global higher-order connectivity patterns of drug nodes in different hypergraphs can be obtained by hierarchically maximizing the mutual information between the representation of the drug, the corresponding drug-centric sub-hypergraph and the hypergraph in each branch. For each motifs-driven hypergraph, an adjacency matrix $\mathbf{I}$ is built to capture the higher-order association information of the drug, with each row representing a drug-centered sub-hypergraph by row-index of the corresponding hypergraph. The sub-hypergraph representation can be obtained by a readout function $\mathcal{F}_s : \mathbb{R}^{s \times d}$, which is depicted as following:

$$
\begin{cases}
\mathcal{D}^x = \mathcal{D} \odot \sigma(\mathcal{D}\mathbf{P}^x_{sg} + \mathbf{b}^x_{sg}) \\
\mathbf{s}^x_d = \mathcal{F}_s(\mathcal{D}^x, \mathbf{q}^x_d) = \dfrac{\mathcal{D}^x \mathbf{q}^x_d}{num(\mathbf{q}^x_d)}
\end{cases}
\tag{9}
$$

where $\mathbf{q}^x_d$ is the row vector of $\mathbf{I}^x$ corresponding to the center drug $d$, and $num(\mathbf{q}^x_d)$ denotes the number of drug $d$ associated nodes in the sub-hypergraph. The representation $\mathbf{s}^x_d$ of the Sub-hypergraph can represent the importance of each drug in the Sub-hypergraph. The hypergraph representation can be obtained by an average pooling to summarize the sub-hypergraph features into a graph level:

$$
\mathbf{g}^x = AveragePooling(\mathbf{S}^x)
\tag{10}
$$

where $\mathbf{S}^x = \{\mathbf{s}^x_1, \mathbf{s}^x_2, ..., \mathbf{s}^x_b\}$.

The pairwise ranking loss [72] is applied to estimate the mutual information, and the objective function of the self-supervised auxiliary task is defined as:

$$
\begin{aligned}
L_{self} = \quad -\sum_{x \in \{j,p,i,u\}} &\left\{ \sum_{d \in D} log\sigma(f(\mathbf{d}^x_d, \mathbf{s}^x_d) - f(\mathbf{d}^x_d, \tilde{\mathbf{s}}^x_d)) \right. \\
&\left. + \sum_{d \in D} log\sigma(f(\mathbf{s}^x_d, \mathbf{g}^x) - f(\tilde{\mathbf{s}}^x_d, \mathbf{g}^x)) \right\}
\end{aligned}
$$

where $D$ is the set of drugs, $f : \mathbb{R}^c \times \mathbb{R}^c \mapsto \mathbb{R}$ denotes a discriminator function with two vectors as the input and scores the agreement between them, and $\tilde{\mathbf{S}}^x = \{\tilde{\mathbf{s}}^x_1, \tilde{\mathbf{s}}^x_2, ..., \tilde{\mathbf{s}}^x_b\}$ is the negative examples by both row-wise and column-wise shuffling to corrupt $\mathbf{S}^x$.

Finally, the overall objective of the model is defined as:

$$
L = L_s + \lambda L_{self}
\tag{12}
$$

where $\lambda$ is a hyper-parameter used to control the effect of the auxiliary task.

## Supporting information

**S1 Text. Fig A. Dimensionality reduction of the drug representations learned from the 4 hypergraph branches**. The figure is drawn by t-SNE and the color corresponds to different types of drug's ATC primary codes. **Table A. Abbreviation List.Table B. The prediction results of other evaluation indicators for HGDrug and three GNN baseline models on four**

**drug interaction datasets**. The best performance is marked in bold and the second best is underlined to facilitate reading. **Table C. Ablation experiments explore the contribution of the four motifs related to fragments.Table D. The top 20 novel drug-drug similarity pairs of the 5 drugs from the DDiI task' drug feature.Table E. The top 20 novel DDIs description details.Table F. The top 20 novel DDIs from HGDrug 2f.Table G. The top 20 novel DDIs from HGDrug 2m.Table H. The top 10 novel targets of paclitaxel.Table I. The top 10 novel side effects of paclitaxel**.
(PDF)

## Author Contributions

**Conceptualization:** Shuting Jin, Xiangrong Liu.

**Data curation:** Shuting Jin, Yinghui Jiang.

**Formal analysis:** Shuting Jin.

**Funding acquisition:** Xiangrong Liu.

**Investigation:** Shuting Jin, Yue Hong.

**Methodology:** Shuting Jin, Yue Hong, Li Zeng.

**Resources:** Shuting Jin, Li Zeng, Yinghui Jiang, Yuan Lin.

**Software:** Shuting Jin, Yue Hong.

**Supervision:** Yuan Lin, Leyi Wei, Xiangxiang Zeng.

**Validation:** Shuting Jin, Yue Hong, Zhuohang Yu.

**Visualization:** Shuting Jin, Yue Hong.

**Writing – original draft:** Shuting Jin, Yuan Lin, Leyi Wei, Xiangxiang Zeng.

**Writing – review & editing:** Shuting Jin, Yuan Lin, Xiangxiang Zeng, Xiangrong Liu.

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
