## [Decision Letter · Decision Letter 0]

25 Aug 2023

Dear Miss Jin,

Thank you very much for submitting your manuscript "A general hypergraph learning algorithm for drug multi-task predictions in micro-to-macro biomedical networks" for consideration at PLOS Computational Biology.

As with all papers reviewed by the journal, your manuscript was reviewed by members of the editorial board and by several independent reviewers. In light of the reviews (below this email), we would like to invite the resubmission of a significantly-revised version that takes into account the reviewers' comments.

We cannot make any decision about publication until we have seen the revised manuscript and your response to the reviewers' comments. Your revised manuscript is also likely to be sent to reviewers for further evaluation.

Sincerely,

Alexander MacKerell

Academic Editor

PLOS Computational Biology

Jason Papin

Editor-in-Chief

PLOS Computational Biology

Reviewer's Responses to Questions

**Comments to the Authors:**

Reviewer #1: Jin et al. reported in this manuscript a hypergraph-based deep learning framework tailed for drug-related predictions using fragments of drug molecules. In particular, BRICSdecomposition is used to partition each small-molecule drug into functional group fragments at a variety of levels, and the interference is performed based on fragments instead of whole molecules. Most importantly, the deployment of hypergraphs by the authors is suitable and imperative for accommodating the higher-order relationships between fragments/motifs. The authors demonstrated that such an approach can be successfully used to perform interference for drug-drug interactions, drug-target interactions, drug-disease relationships, and drug side-effect interactions. The manuscript is in general well-written, and should be suitable for PLOS Comp. Biol. It can be improved by addressing the following comments.

1) The section discussing implementation specifics should elucidate the number of learnable parameters in the neural network (NN) models. This should be in tandem with the information about training parameters and the utilized platform.

2) To ensure a fair comparison, the inclusion of other evaluation metrics—such as MCC, Precision, Recall, and F1-score—is advised for all methods in comparison across all four drug-related tasks.

3) The manuscript would benefit from a more comprehensive description of the baseline models, especially concerning the three GNN-based models. Given that the efficacy of NNs is closely tied to their configurations, the authors, at the very least, should specify the number of learnable weights in the baseline models in relation to their HGDrug models.

4) I checked the corresponding github repository and found the introduction unclear. It would be beneficial to incorporate a comprehensive README section, detailing download instructions, software requirements, training protocols, and illustrative examples of outputs.

5) Table 1 requires a more detailed explanation and description.

Besides, I'd like to draw attention to a potential typo error concerning the eighth affiliation of the corresponding author (X.L.). Please check whether it is "Zhejiang Lab" as opposed to "Zhijiang Lab".

Reviewer #2: Summary:

The authors present a hypergraph learning algorithm named HGDrug for drug multi-task predictions. The algorithm incorporates drug-substructure relationships into molecular interaction networks to construct a drug-centric heterogeneous network. HGDrug captures high-order drug relations and fetches effective drug features using motif-driven hypergraphs and a self-supervised auxiliary task. This study demonstrates HGDrug achieves state-of-the-art performance and the ability to capture relations between drugs with the same functional groups. The proposed drug-substructure interaction networks can also improve the performance of existing network models for drug-related prediction tasks. The paper is well-written, and the method is clearly described. I only have a few comments and suggestions for the authors.

Major:

The introduction does provide a clear objective of the study, which is to introduce a new hypergraph learning algorithm for drug multi-task predictions. However, the significance of this objective in the broader context of drug discovery could be elaborated upon more.

I suggest they give a deeper explanation about why their objective is challenged or crucial.

While the work does touch upon the importance of drug multi-task predictions, a more detailed motivation explaining the challenges in the current landscape would provide a stronger foundation for the study.

The authors briefly mention terms like "motif-driven hypergraphs" and "self-supervised auxiliary task". It would be better to give a brief explanation or reference for these terms in the introduction.

While the paper claims to introduce a novel approach by incorporating drug substructure information, it would be beneficial to provide a clearer distinction description between the proposed method and existing graph-based methods. For example, give more information to introduce the difference between DSMN and the heterogeneous-network-based methods.

I suggest the authors underscore the significance and benefits of their approach compared to traditional methods. Many researchers tend to favor established techniques unless the new method addresses specific challenges that traditional methods cannot overcome.

Why is there a pressing need for another drug prediction algorithm?

What specific challenges does HGDrug address that others don't?

The work could benefit from better structuring, possibly with more defined subheadings, to guide the reader through different sections. I suggest the authors could summarize the section's conclusion as the subheading for readers’ better understanding.

It would be beneficial to provide a detailed explanation for the operator “δ||θ|_2^2” in Formula 8.

For discussion

It would be better to highlight to emphasize the importance of the potential applications or implications of the HGDrug algorithm in real-world scenarios could be.

I suggest the authors have a discussion on the computational complexity of HGDrug, its scalability, and performance concerning dataset size would provide insights into its feasibility for large-scale applications.

Minor:

Refining and condensing your writing will improve the paper's readability. For example: In the Definition 2 section, "The motif is a small pattern of interconnections occurring in complex networks at numbers that are significantly higher than those in randomized networks," consider revising to "A motif is a recurring pattern of interconnections in complex networks, occurring more frequently than in randomized networks."

For references, ensure all references are up-to-date and relevant. Cross-check that all cited works are appropriately referenced within the text.

In this article, there are many abbreviations, it would be better to consider providing a list or table of abbreviations for quick reference.

The manuscript delves into technical details regarding hypergraphs, motifs, and heterogeneous networks. It would be helpful to provide more intuitive explanations or examples for readers less familiar with these concepts.

Reviewer #3: The manuscript describes a hypergraph learning model, namely HGDrug, which introduces drug substructure (functional group fragments) into biomedical networks for the first time. Specifically, the work constructs a drug-centric micro-to-macro heterogeneous network (termed DSMN) and presents a motif-driven hypergraph learning framework (termed HGDrug) with the self-supervised auxiliary task for drug multi-task interaction predictions. The method is evaluated by 4 benchmark tasks and demonstrates that it achieves highly accurate and robust predictions, outperforming 8 state-of-the-art task-specific models and 6 general-purpose conventional models. A specific case study also shows that HGDrug can learn the substructure information to improve the performance of drug repurposing. However, I have the following concerns.

Major comments:

1、 Fragments increase the number of nodes and edges of these networks. For example, for the Drug-drug, Drug-target, Drug-disease, and Drug-side-effect datasets, the number of fragments is 4.42 times the number of nodes. Please discuss the balance between time and performance before and after adding fragments.

2、 In the part of the “Ablation studies”, the author explored the influence of 4 hypergraph-based branches on the prediction performance of the model, but I noticed that each hypergraph branch also contains multiple network motifs to derive hypergraphs. I think the author should study the contribution of the four kinds of motifs (M2, M3, M5, M6) related to fragments to the prediction results, and further verify the importance of constructing hypergraphs based on fragments for drug feature learning.

3、 The work uses BRICS to decompose the SMILES sequences of drugs when acquiring DFIs and FFIs networks. As far as I know, there are many ways for SMILES decomposition, such as rdkit Recap, why did the author choose BRICS? BRICSDecompose should return a non-repeated fragments list after decomposition. How did the author achieve FFI?

4、 At the beginning of the Section “Network visualization of the DDiI predictions”, it is not clear what the authors mean by " We remove the known DDiIs used in the prediction model…". Do they perform again the same pipeline without considering the known drug-disease associations? Do they delete the known associations from the outcome they've already had in Section ‘Performance comparison’?

Minor comments:

1、In the results shown in Figure 7, there seems to be no“Disease/Side effect” node type, and the color marked in the figure seems to be the same as the “Target” node.

2、 Some important works have not been reviewed or mentioned, such as doi: 10.1093/bioinformatics/btac579, doi: 10.1093/bioinformatics/btab651, doi: 10.1093/bioinformatics/btz718，doi：10.1371/journal.pcbi.1011382. Sufficient literature is important for potential readers.

**Have the authors made all data and (if applicable) computational code underlying the findings in their manuscript fully available?**

Reviewer #1: Yes

Reviewer #2: Yes

Reviewer #3: Yes

PLOS authors have the option to publish the peer review history of their article (what does this mean?). If published, this will include your full peer review and any attached files.

Reviewer #1: **Yes: **Jing Huang

Reviewer #2: No

Reviewer #3: No
---

## [Decision Letter · Decision Letter 1]

13 Oct 2023

Dear Miss Jin,

We are pleased to inform you that your manuscript 'A general hypergraph learning algorithm for drug multi-task predictions in micro-to-macro biomedical networks' has been provisionally accepted for publication in PLOS Computational Biology.

Best regards,

Alexander MacKerell

Academic Editor

PLOS Computational Biology

Jason Papin

Editor-in-Chief

PLOS Computational Biology

Reviewer's Responses to Questions

**Comments to the Authors:**

Reviewer #1: The authors have adequately addressed my comments and the manuscript is now ready for publication.

Reviewer #3: The authors have addressed my concerns.

**Have the authors made all data and (if applicable) computational code underlying the findings in their manuscript fully available?**

Reviewer #1: Yes

Reviewer #3: None

PLOS authors have the option to publish the peer review history of their article (what does this mean?). If published, this will include your full peer review and any attached files.

Reviewer #1: **Yes: **Jing Huang

Reviewer #3: No

---

## [Editor Report · Acceptance letter]

2 Nov 2023

PCOMPBIOL-D-23-01067R1 

A general hypergraph learning algorithm for drug multi-task predictions in micro-to-macro biomedical networks

Dear Dr Jin,

I am pleased to inform you that your manuscript has been formally accepted for publication in PLOS Computational Biology. Your manuscript is now with our production department and you will be notified of the publication date in due course.

With kind regards,

Anita Estes
